# Importance of ACE2 for SARS-CoV-2 Infection of Kidney Cells

**DOI:** 10.3390/biom13030472

**Published:** 2023-03-03

**Authors:** Marie-Kristin Kroll, Sebastian Schloer, Peynaz Candan, Nadia Korthals, Christoph Wenzel, Hannah Ihle, Kevin Gilhaus, Kim Rouven Liedtke, Michael Schöfbänker, Beate Surmann, Rita Schröter, Ute Neugebauer, Gita Mall, Stefan Oswald, Stephan Ludwig, Ursula Rescher, Beate Vollenbröker, Giuliano Ciarimboli

**Affiliations:** 1Medizinische Klinik D, Universitätsklinikum Münster, 48149 Münster, Germany; 2Research Group Regulatory Mechanisms of Inflammation, Institute of Medical Biochemistry, Centre for Molecular Biology of Inflammation, University of Münster, 48149 Münster, Germany; 3Leibniz Institute of Virology, 20251 Hamburg, Germany; 4Institut für Pharmakologie, Universitätsmedizin Greifswald, 17489 Greifswald, Germany; 5Institute of Forensic Medicine, University Hospital Jena, 07747 Jena, Germany; 6Department of General, Visceral, Thoracic and Vascular Surgery, Greifswald University Medical Center, 17475 Greifswald, Germany; 7Institute of Virology (IVM), Center for Molecular Biology of Inflammation (ZMBE), University Hospital Münster, University of Münster, 48149 Münster, Germany; 8Institut für Pharmakologie und Toxikologie, Universitätsmedizin Rostock, 18057 Rostock, Germany

**Keywords:** SARS-CoV-2, kidneys, ACE2

## Abstract

In late 2019, the novel severe acute respiratory syndrome coronavirus 2 (SARS-CoV-2) as the causative agent of coronavirus disease 2019 (COVID-19) emerged in China and spread rapidly around the world, causing an ongoing pandemic of global concern. COVID-19 proceeds with moderate symptoms in most patients, whereas others experience serious respiratory illness that requires intensive care treatment and may end in death. The severity of COVID-19 is linked to several risk factors including male sex, comorbidities, and advanced age. Apart from respiratory complications, further impairments by COVID-19 affecting other tissues of the human body are observed. In this respect, the human kidney is one of the most frequently affected extrapulmonary organs and acute kidney injury (AKI) is known as a direct or indirect complication of SARS-CoV-2 infection. The aim of this work was to investigate the importance of the protein angiotensin-converting enzyme 2 (ACE2) for a possible cell entry of SARS-CoV-2 into human kidney cells. First, the expression of the cellular receptor ACE2 was demonstrated to be decisive for viral SARS-CoV-2 cell entry in human AB8 podocytes, whereas the presence of the transmembrane protease serine 2 (TMPRSS2) was dispensable. Moreover, the ACE2 protein amount was well detectable by mass spectrometry analysis in human kidneys, while TMPRSS2 could be detected only in a few samples. Additionally, a negative correlation of the ACE2 protein abundance to male sex and elderly aged females in human kidney tissues was demonstrated in this work. Last, the possibility of a direct infection of kidney tubular renal structures by SARS-CoV-2 was demonstrated.

## 1. Introduction

In December 2019 in Wuhan, China, a cluster of pneumonia emerged, caused by infection with the severe acute respiratory syndrome coronavirus type 2 (SARS-CoV-2), which represented the beginning of the pandemic coronavirus disease 2019 (COVID-19) [1]. SARS-CoV-2, the etiological agent of COVID-19, belongs to the coronaviruses (CoVs). CoVs are non-segmented, enveloped viruses with variable size (80–120 nm in diameter) which contain a plus stranded RNA genome of 27–32 kb [2,3]. The genome of CoVs encodes several non-structural proteins, important for RNA synthesis and processing, as well as structural proteins (the transmembrane M- and the spike S-glycoprotein, the nucleocapsid N- and the envelope E-protein) [3,4,5]. The S-protein forms large, trimeric spikes which confer the virus its typical, crown-shaped appearance [6]. These large spikes induce virus–cell fusion by mediating attachment and fusion of the virus to its target cell [3]. For viral cell entry, a defined receptor-binding domain (RBD) within the N-terminal S1 subunit of the S-protein binds to a cellular receptor on the surface of the host cell, whereas the C-terminal S2 subunit anchors the S-protein to the cellular membrane and mediates membrane fusion. The monocarboxypeptidase angiotensin-converting enzyme 2 (ACE2) is described as the major cell entry receptor of SARS-CoV-2 [7,8,9,10]. In addition to ACE2, further receptors, such as the transmembrane glycoprotein Neuropilin-1 (NRP1) [11] or the liver/lymph node-specific ICAM-3-grabbing non-integrin (CD209L) [12], are discussed to drive or potentiate the cellular uptake of SARS-CoV-2. 

In addition to binding of the viral spike protein to a receptor on the surface of a target cell, the activity of the plasma-anchored transmembrane protease serine 2 (TMPRSS2) seems to be important for cleaving the viral spike protein at defined sites, evoking a conformational change and activating the S-protein for membrane fusion [8]. 

The manifestations of COVID-19 are very heterogeneous. A large part of infections proceeds without or with only mild flu-like symptoms, whereas some patients exhibit severe symptoms including high fever and an acute respiratory distress syndrome (ARDS) as a major complication [13,14,15]. ARDS requires intensive care including oxygen treatment or artificial respiration and may end in death [16]. In addition to primarily affecting the lungs, COVID-19 is associated with several extrapulmonary complications including for example coagulopathy, which manifests as thrombosis in different tissues and is linked to poor prognosis [17,18], cardiovascular impairments such as arrhythmias, myocardial injury, or myocarditis [13,19], neurological manifestations [20,21], and gastrointestinal symptoms [10,15]. As a frequent extrapulmonary complication, acute kidney injury (AKI) may develop, which is associated with higher case fatality rates [10,22]. Thereby, kidney injury is considered to be a clinical indicator of severe COVID-19 progression [23]. A cytokine storm (cytokine release syndrome—CRS) induced by viral infection seems to be one cause of renal damage in COVID-19 patients [23]. However, kidneys might be directly infected by SARS-CoV-2 [23]. Renal SARS-CoV-2 infection requires the presence of a virus receptor (e.g., ACE2) in the kidneys. As virus binding to a receptor goes along with internalization of the receptor-virus complex, an interaction with renal ACE2 might not only be important for virus infection, but consequently also for imbalance of ACE2 signaling. ACE2 is a single-pass transmembrane glycoprotein [24] consisting of an intracellular C-terminal membrane anchor domain and an extracellular N-terminal domain, which exhibits an active catalytic site and serves as cellular receptor. Physiologically, ACE2 functions as a carboxypeptidase which removes phenylalanine at the C-terminus of Angiotensin II (Ang II), resulting in the formation of Ang (1–7). ACE2 is also able to form Ang (1–9) from Angiotensin I (Ang I). Because of this function, ACE2 is an essential component of the renin–angiotensin system (RAS); see Figure 1. 

The RAS reaction cascade plays a crucial role in the regulation of blood pressure and the maintenance of electrolyte and water balance [26]. In this cascade, renin cleaves Angiotensinogen into the decapeptide Ang I, which is then converted by Angiotensin-converting enzyme (ACE) into the vasoactive octapeptide Ang II [27]. The so-formed Ang II binds and activates the angiotensin II receptor type 1 and 2 (AT1R, AT2R). Binding to AT1R induces well-known physiological effects of Ang II such as vasoconstriction, while binding to AT2R is supposed to have a counteracting effect. In general, activation of these receptors induces either harmful events such as inflammation, coagulation, and fibrosis (AT1R) or counterbalancing events including anti-fibrotic, anti-coagulatory, anti-inflammatory, or metabolic functions (AT2R) [25]. 

ACE2-cleaving of Ang I to Ang (1–9) and of Ang II to Ang (1–7), which subsequently interacts with the MAS-Receptor (Mas-R), is important for the RAS protecting and regenerating pathway (Figure 1) [28,29]. An imbalance of the RAS system towards Ang II has been observed in many pathological conditions such as myocardial infarction, heart failure, hypertension, or renal diseases. Indeed, such pathological conditions benefit from treatment using ACE inhibitors or AT1R blockers [27]. Since the ACE2/SARS-CoV-2 complex is endocytosed after virus binding to ACE2, allowing the virus to enter target cells, a decrease in ACE2 activity is expected upon viral infection. This may increase the activity of the “deleterious” arm of RAS locally and/or systemically. In fact, such a possibility is intensively debated (see for example [30]). 

The severity of COVID-19 is associated with several risk factors. Even though total infectivity rates of male and female patients are comparable, the rate of severely affected and hospitalized patients is significantly higher in men compared to women [15,31,32]. In this regard, Karagiannidis et al. reported nearly twice the rate of male patients requiring ventilation compared to females [33]. This sex-associated severity of disease courses results in significantly higher fatality rates in men compared to women [34]. Moreover, age is described as another factor favoring fatal infection progression. Indeed, infections in pediatric age are reported, but serious COVID-19 courses in children are described as exceptions [35,36]. For all SARS-CoV-2 variants, even children admitted to intensive care units (ICUs) generally recovered without any adverse outcomes [37]. Conversely, older people have a higher risk of unfavorable progression and increased mortality rates. Consequently, there is a particularly increased risk of severe infection in older men due to sex specificity [15]. As additional risk factors for a serious course of COVID-19 and the necessity of ICU treatment, a wide range of pre-existing conditions including diabetes mellitus, obesity, hypertension, or cardiovascular diseases, for example, are described [32,33,38].

Further, the impairment of renal function in COVID-19 could be not only a direct consequence of viral cell entry but also a secondary consequence of viral infection initially leading to immune system or coagulation cascade dysfunction. For example, AKI and renal failure are reported in association with cytokine release syndrome (CRS) [39]. Apart from renal failure, also cardiac impairment and respiratory symptoms possibly ending in ARDS are known effects of CRS [39].

This study aims to investigate mechanisms of the SARS-CoV-2 cell entry process into kidney cells, especially focusing on the cell entry receptor ACE2 and the S-protein priming protease TMPRSS2. Protein expression patterns of ACE2 and TMPRSS2 in human kidneys and their dependence on sex and age were investigated. Finally, direct SARS-CoV-2 infection of renal structures was examined in autoptic kidney samples from deceased COVID-19 patients.

## 2. Materials and Methods

### 2.1. Human Samples

Human kidney samples were obtained from Caucasian male and female patients undergoing tumor nephrectomy in the Clinic for Urology of the University Hospital Münster (ethics-vote 2020-858-f-S Ethics committee of University Hospital Münster, Germany, 19 February 2021). Immediately after nephrectomy, a piece of normal kidney tissue far away from the tumor and without any signs of necrosis was withdrawn and transferred into chilled HCO_3_^−^–free phosphate buffer. Colon samples were collected by the Department of Clinical Pharmacology of the University Medicine Greifswald (ethics-vote BB222/20 Ethics committee University Hospital Greifswald, Germany). Samples were collected from patients undergoing intestinal surgeries but showing no signs of inflammation or necrosis. Written consent was obtained from all patients. No patient was affected by COVID-19. All samples were stored at −80 °C. Paraffin-embedded kidney samples from deceased COVID-19 patients were collected and kindly provided by the University Hospital of Jena, Germany (ethics-vote 2020-1773 Ethics committee University Hospital Jena).

### 2.2. Protein Quantification by LC-MS/MS

Protein abundance of ACE2, TMPRSS2, and the reference protein Na^+^/K^+^-ATPase in membrane protein fractions from kidney and colon samples were determined using a validated LC-MS/MS-based targeted proteomics assay [40]. The principle of this method is based on the quantification of proteospecific peptides, which are released during tryptic digestion, and allows determination of absolute protein abundance.

The membrane protein fractions from tissue samples were extracted using the ProteoExtract Native Membrane Protein Extraction kit (Merck, Darmstadt, Germany) according to the manufacturer’s protocol with slight modifications. In brief, frozen and powdered tissue samples were suspended in 500 µL cell lysing extraction buffer I (containing 5 µL/mL protease inhibitor cocktail) and homogenized with a Dounce homogenizer before being incubated for 15 min at 4 °C while shaking. Afterwards the homogenates were centrifuged at 16,000× *g* for 15 min at 4 °C. The supernatants, containing cytosolic proteins, were discarded and the received pellets were resuspended in 300 µL extraction buffer II (containing 5 µL/mL protease inhibitor cocktail). After incubation for 60 min at 4 °C while shaking, the suspensions were again centrifuged for 15 min at 16,000× *g* and 4 °C. The supernatants containing membrane proteins were collected. Total protein concentrations of the obtained membrane fractions were determined with the bicinchoninic acid assay (Thermo Fisher Scientific, Schwerte, Germany). If necessary, membrane fractions were adjusted to a maximum protein concentration of 2 mg/mL. Subsequently, 100 µL of each membrane fraction was mixed with 10 µL dithiothreitol (200 mM, Sigma-Aldrich, Taufkirchen, Germany), 40 µL (NH_4_)HCO_3_ buffer (50 mM, pH 7.8, Sigma-Aldrich, Taufkirchen, Germany), and 10 µL ProteaseMAX™ (1%, m/v, Promega, Mannheim, Germany) and incubated for 30 min at 60 °C (denaturation). After cooling down, 10 µL iodoacetamide (400 mM, Sigma-Aldrich, Taufkirchen, Germany) was added and the samples were incubated in a darkened water quench for 15 min at 37 °C (alkylation). For protein digestion, 10 µL trypsin (trypsin/protein ratio: 1/40, Promega, Mannheim, Germany) was added and the samples were incubated in a water quench for 16 h at 37 °C. Digestion was stopped by addition of 20 µL formic acid (10% *v*/*v*, Thermo Fisher Scientific, Schwerte, Germany). All samples were stored at −80 °C until further processing. Before measurements, samples were centrifuged one more time for 15 min at 16,000× *g* and 4 °C. Finally, 50 µL of the supernatant was mixed with 50 µL isotope-labeled internal standard (IS) peptide mix (10 nM of each IS). For preparation of the calibration curves, digested human serum albumin (2 mg/mL) was used as blank matrix and spiked with reference peptides to reach 0.1–25 nM and with an isotope-labeled internal standard (final concentration 10 nM of each peptide). The following proteospecific peptides have been used: ACE2: GDYEVNGVDGYDYSR; TMRPSS2: LNTSAGNVDIYK; and Na^+^/K^+^-ATPase: LSLDELHR. Peptides were selected as recently described [41]. All peptides were synthesized on demand in high analytical quality (purity >97–99%) by Thermo Fisher Scientific (Thermo Fisher Scientific, Schwerte, Germany).

All sample preparation and digestion steps were performed using Protein LoBind tubes (Eppendorf, Hamburg, Germany). Protein quantification was conducted on a 5500 QTRAP triple quadrupole mass spectrometer (AB Sciex, Darmstadt, Germany) coupled to an Agilent Technologies 1260 Infinity system (Agilent Technologies, Santa Clara, CA, USA).

For each peptide, 3–4 mass transitions were monitored and the absolute protein abundance was assessed by using the stable isotope method and considering the protein content of the sample (data given as pmol target protein per mg membrane protein). The following mass transitions were used for quantification (Q1 and respective Q3 mass to charge ratios): ACE2 (GDYEVNGVDGYDYSR): 855.3/875.3, 465.4, 564.2, and 760.4; TMRPSS2 (LNTSAGNVDIYK): 648.2/228.3, 310.2, 808.7, and 879.8; and Na^+^/K^+^-ATPase (LSLDELHR): 328.3/435.2, 391.7, and 669.3. Details of the used chromatographic method were described elsewhere [40].

### 2.3. Immunofluorescence Staining

For immunofluorescence staining in different cell lines, cells were cultured on glass coverslips up to a confluence of 70–90%. Cell culture medium was removed, and cells were rinsed with ice-cold phosphate buffered saline (PBS) twice. For subsequent fixation of the cells, coverslips were incubated in 4% paraformaldehyde (PFA) for 20 min at room temperature. Afterwards, cells were washed with PBS for 10 min on a shaking incubator for three times. To block free aldehydes, a quenching process was performed using 50 mM NH_4_Cl in 1x PBS for 5 min. After three further washing steps, permeabilization of cells in 0.2% Triton-X 100 in PBS followed. Thereafter, three further washing steps using immunofluorescence washing solution (IF WS: 10% (*v*/*v*) 10 x PBS, 0.2% (*w*/*v*) gelatin, 0.2% (*v*/*v*) Triton X-100 in distilled water) and blocking in immunofluorescence blocking solution (BS: 90 % (*v*/*v*) IF WS, 10% (*v*/*v*) normal goat serum (NGS)) for 20 min to prevent non-specific antibody bindings were performed. For the detection of proteins, primary antibodies were diluted in the desired concentration in antibody solution (98% (*v*/*v*) IF WS, 2% (*v*/*v*) NGS) and incubated overnight at room temperature in a wet chamber. The next day, primary antibodies were removed by three washing steps with IF WS, and incubation with secondary antibodies ensued. To additionally visualize cell nuclei, 4′,6-diamidino-2-phenylindole (DAPI, 1:5000) was added to the secondary antibody solution and incubated for 1 h in the dark. After incubation and three times rinsing with PBS, coverslips were washed in distilled water for 20 s and mounted on microscope slides with Mowiol (Carl Roth, Karlsruhe, Germany). Samples were stored at 4 °C in the dark.

Localization of ACE2 and TMPRSS2 in human kidneys was investigated using specific antibodies. Cryosections (5 µm) were permeabilized with 0.2% Triton X-100 in PBS for 3 min, washed with PBS and then incubated for 1 h at room temperature in PBS containing 10% BSA (pH 7.4). After this, sections were incubated overnight at room temperature with primary antibodies against ACE2 (Cell Signaling Technology, Inc., Danvers, MA, USA), TMPRSS2 (Abcam, Cambridge, UK) or CD209L (Sigma-Aldrich, Taufkirchen, Germany) at a dilution of 1:100. Proximal and distal tubules were identified by simultaneous staining with phytohaemagglutinin (PHE-A, 1:200 in PBS, green fluorescence) or peanut agglutinin (PNA, 1:100 in PBS, red fluorescence), respectively (both Biologo, Kronshagen, Germany) [42].

After washing in PBS, cryosections were incubated for 1 h at room temperature with secondary antibodies (Alexa Fluor 647 donkey anti-goat IgG, Invitrogen (Thermo Fisher Scientific, Schwerte, Germany), 1:1000) diluted in PBS, rinsed with PBS, and then incubated with DAPI in PBS (1:5000) for 2 min, mounted on microscope slides with Mowiol (Carl Roth, Karlsruhe, Germany) and evaluated by epifluorescence microscopy (Observer Z1 with Apotome, Carl Zeiss, Oberkochen, Deutschland). Negative control slides were included without the addition of primary antibodies.

### 2.4. Immunohistochemistry

Immunohistochemistry (IHC) experiments were performed on paraffin-embedded kidney tissues to selectively localize proteins and investigate tissue morphology. For IHC, samples were prepared by melting paraffin at 72 °C for 30 min in an incubator. Afterwards, samples were incubated in xylol for 10 min twice and irrigated by a descending alcohol series of 99%, 96%, and 70% ethanol for 3 min each. Next, heat pre-treatment was performed by boiling the slices at 120 °C for 5 min in TRIS-EDTA buffer and subsequent incubation in distilled water. Afterwards tissues were prepared using the REAL EnVision Detection System kit by Dako (Dako, Hamburg, Germany) following the manufacturer’s instruction. Additionally, haematoxylin staining was performed for 5 min at room temperature to stain cell nuclei followed by a 5 min washing step under tap water with an associated pH change to stop the reaction. Finally, samples were covered with coverslips and stored at room temperature. For SARS-CoV-2 detection in autoptic kidney samples from COVID-19 deceased patients, all reaction steps were performed using the VENTANA BenchMark ULTRA system by Roche (Roche Applied Science, Mannheim, Germany). In this apparatus, paraffin was removed at 72 °C and heat treatment was performed at 95 °C for 40 min with the Cc1 (Cell conditioning 1) buffer (Roche Applied Science, Mannheim, Germany). Next, the SARS-CoV-2 nucleocapsid primary antibody (40143-MM05 Sino Biological Europe, Eschborn, Germany) diluted to 1:250 was incubated for 32 min at 36 °C and, subsequently, the OptiView DAB Detection Kit (Roche Applied Science, Mannheim, Germany) was used. Finally, haematoxylin staining (haematoxylin II, Roche Applied Science, Mannheim, Germany) was conducted, and samples were incubated with Blueing Reagent (Roche Applied Science, Mannheim, Germany) for 5 min. Slides were stored at room temperature.

### 2.5. Generation of Stable Cell Lines

For stable lentiviral transduction, HEK293T cells (ATCC CRL- 1573) were simultaneously transfected with 6.5 µg psPAx2, 3.5 μg pMDs-VSVG (both gifts from D. Trono, Addgene plasmids #12260 and #12259, respectively), and 10 μg pWPI-IRES-Puro-Ak-ACE2, pWPI-IRES-Puro-Ak-TMPRSS2, or pWPI-IRES-Puro-Ak ACE2+TMPRSS2 (these plasmids were a gift from S. Best: Addgene plasmids #154985, #154986, and #154987, respectively) by calcium phosphate method, as described in [43]. The medium of transiently transfected HEK293T cells was changed after 6–8 h and cells were grown for an additional 72 h. After that time, the virus-containing supernatant was collected and filtered through a sterile 0.45 μm syringe driven filter unit (Millipore, Schwalbach am Taunus, Germany). Subsequently, human immortalized podocytes (AB8, kindly provided by M. Saleem [44]) growing in 6-well dishes were infected for 24 h using one volume (up to 2 mL) of fresh podocytes medium (standard RPMI-1640 medium (Sigma-Aldrich, Taufkirchen, Germany) containing 100 U/mL penicillin, 100 µg/mL streptomycin, 10% fetal bovine serum (Biochrom, Berlin, Germany), and supplements, as described in [45]) and one volume of the virus-containing filtrate supplemented with polybrene (final concentration 8 μg/mL). Thereafter, the virus-containing medium was replaced by fresh medium and cells were regenerated for 24 h. Successfully transduced cells were selected after a second regeneration period by puromycin (2 μg/mL). Overexpression of the target proteins was verified by PCR- and Western blot-analysis.

### 2.6. Real-Time PCR Analysis

Cells were washed, and total RNA was isolated using the GenElute Mammalian Total RNA Miniprep Kit (Sigma-Aldrich, Taufkirchen, Germany). For cDNA synthesis, 2 µg total RNA was used with the LunaScript SuperMix Kit (New England BioLabs, Ipswich, MA, USA). Gene expression profiles for ACE2, TMPRSS2, CD209, CD209-L, and Neuropilin-1 (NRP1) were analyzed by real time PCR using specific primer pairs as listed in Table 1 and SYBR Select Master Mix for CFX (Thermo Fisher, Waltham, MA, USA) on a CFX Realtime Detection System (Biorad, Hercules, CA, USA). Relative gene expression values were evaluated with the 2^-ΔCt^ method [46] using GAPDH as housekeeping gene.

### 2.7. Infection Assay

Infection with the SARS-CoV-2 isolates South Tyrol (hCoV-19/Germany/FI1103201/2020) and with the Delta variant B.1.617.2 from the strain collection of the Institute of Virology, University of Münster, Germany, was carried out as previously described [47]. Briefly, cells were washed with PBS and infected in infection-PBS (containing 0.2% BSA, 1% CaCl_2_, 1% MgCl_2_, 100 U/mL penicillin, and 0.1 mg/mL streptomycin) at a multiplicity of infection (MOI) of 0.1 for 1 h. Afterwards, cells were washed twice with PBS and cultured in infection-medium (RPMI 1640 medium containing 2% fetal calf serum, 100 U/mL penicillin, and 0.1 mg/mL streptomycin) for a total infection period of 48 h.

### 2.8. Plaque Assay

To quantify viral titers in the supernatants of infected cells, a standard plaque assay was performed [48]. In brief, Vero E6 cells grown to a monolayer in six-well dishes were infected with serial dilutions of the respective supernatants in infection-PBS for 1 h at 37 °C. The inoculum was replaced with 2x MEM (MEM containing 0.2% BSA, 2 mM L-glutamine, 1 M HEPES, pH 7.2, 7.5% NaHCO_3_, 100 U/mL penicillin, 0.1 mg/mL streptomycin, and 0.4% Oxoid agar) and incubated at 37 °C. To visualize virus plaques a neutral red solution was used, and virus titers were calculated as plaque-forming units (PFU) per mL.

### 2.9. SARS-CoV-2 Pseudovirus Preparation and Infection

Pseudoviruses presenting the spike (S) proteins of SARS-CoV-2 variants Wuhan, Delta (B.1.617.2), and Omicron (BA.1 and BA.5) were generated using a replication-deficient vesicular stomatitis virus (VSV) bearing the eGFP reporter gene (VSVΔG/GFP, kindly provided by Gert Zimmer, Institute of Virology and Immunology IVI, Mittelhäusern/Switzerland) as previously published [8]. HEK-293T cells transiently transfected with expression plasmids encoding the SARS-CoV-2 S proteins for 18–20 h were inoculated with VSV∆G/GFP transcomplemented with the VSV-G surface protein for 1 h. Subsequently, cells were washed and cultured with supernatant of the I1 hybridoma cells expressing α-VSV-G antibody added to the medium for 30 min to remove remaining VSV-G bearing pseudoviruses. Cells were washed and cultivated further. At 20 h, the supernatants containing the VSV pseudotype particles bearing the SARS-CoV-2-S proteins (VSV∆G/GFP-S_Wuhan_, VSV∆G/GFP-S_Delta_, VSV∆G/GFP-S_Omicron_) were harvested and cellular debris was removed by centrifugation. Pseudoviruses were aliquoted and stored at −80 °C. Cells were inoculated with the pseudotyped VSV bearing the SARS-CoV-2 S proteins at MOI 0.03 for A549 cell line and MOI 1 for AB8 podocytes for 1 h at 37 °C. Cells were then washed and further cultivated. Entry of pseudovirus particles was analyzed 16 h later by counting GFP-positive cells. Both cell lines were analyzed using the Celigo Image Cytometer (Nexcelom Bioscience, Lawrence, MA, USA).

### 2.10. Statistical Analysis

Data were analyzed using GraphPad Prism, Version 9.0 (GraphPad Software, Inc., San Diego, CA, USA). When not otherwise indicated, data presented in this work are expressed as mean values ± SEM, with N referring to the number of independent experiments. Unpaired *t*-tests and/or ANOVA tests with post-hoc analyses were applied to prove statistical significance (*p* < 0.05).

## 3. Results

### 3.1. Localization of Entry Receptors in Human Kidneys

To identify possible SARS-CoV-2 target cells within human kidneys, we investigated the renal expression of the entry receptor ACE2 and of the host protease TMPRSS2. Interestingly, a significant ACE2 expression was observed mainly in the apical membrane domain of renal tubule cells (Figure 2).

To better identify the nephron segment, where ACE2 is localized, we performed ACE2 co-localization experiments with phytohaemagglutinin (PHE-A) and peanut agglutinin (PNA), which are specific markers of the apical domain of proximal and distal renal tubules [42], respectively (Figure 3).

Moreover, using the same approach, TMPRSS2 and CD209L renal distribution was investigated. ACE2 was clearly distributed on the apical membrane domain of proximal tubules (Figure 3a). TMPRSS2 was mainly present in the membrane domain of proximal and distal tubules (Figure 3b). CD209L expression was also evident mainly in renal proximal tubules (Figure 3c), but it showed a less intense staining of the plasma membrane than ACE2 and TMPRSS2.

### 3.2. Role of Virus Entry Receptors for Infection of AB8 Podocytes

Using human immortalized AB8 podocytes, we investigated the dependency of SARS-CoV-2 infection from the presence of ACE2 and/or TMPRSS2. First, we compared the endogenous expression of putative virus receptors in AB8 podocytes with that measured in Caco2 and Calu3 cells, human cell lines of intestinal and respiratory origin, respectively, which are well known to be susceptible to SARS-CoV-2 infection [49,50]. The most striking differences between AB8 podocytes and Caco2 and Calu3 cells is that these two cell lines at least at mRNA level contain much more ACE2 than AB8 podocytes (Figure 4). Caco2 cells also express high amounts of TMPRSS2. The expression level of NRP1 and CD209L was similar in all the cell lines. In AB8 podocytes, the Ct values for mRNA expression of ACE2, TMPRSS2, NRP1, CD209L, and GAPDH (as a loading control) were 30.5 ± 0.1, 27.4 ± 1.0, 23.3 ± 1.3, 29.1 ± 0.4, and 15.1 ± 0.6, respectively, showing that podocytes endogenously express mainly NRP1 (Ct value = 23) and TMPRSS2 (Ct value = 27) at a measurable level.

While Caco2 and Calu3 cells were markedly infected by the B.1.617.2 and FI1103201 SARS-CoV-2 variants, AB8 human podocytes were resistant against virus infection (Figure 5). Therefore, we focused on ACE2 as a possible determinant of SARS-CoV-2 infectivity using AB8 human podocytes.

We compared SARS-CoV-2 infectivity in AB8 podocytes, where the expression of ACE2 or/and TMPRSS2 was increased by viral transduction (Figure 6). The mRNA and protein expression of ACE2 and TMPRSS2 in AB8 podocytes after viral transduction are presented in Appendix A.

Only after ACE2 overexpression, AB8 podocytes could be infected by SARS-CoV-2, suggesting that ACE2 is necessary for virus infection. Overexpression of TMPRSS2 alone did not confer virus infectivity. The combined TMPRSS2/ACE2 overexpression did not further increase virus infectivity compared to overexpression of ACE2 alone, suggesting that sufficient amounts of endogenous TMPRSS2 and/or other TMPRSS2-like proteases were already expressed. Given that ACE2 is necessary for SARS-CoV-2 infection of AB8 podocytes, ACE2 may have a special importance for SARS-CoV-2 infection of kidney cells. To focus on the viral entry process in more detail, we utilized replication-defective VSV particles pseudotyped with the S-proteins of the SARS-CoV-2 variants Wuhan, Delta, and Omicron (Figure 7). With this system, only susceptibility to virus entry is analyzed, as the infection cycle is halted once the viral genome including the GFP reporter gene is transferred into the host cell. Our results confirmed that ACE2 but not TMPRSS2 expression was indispensable for SARS-CoV-2 entry, as no GFP-positive cells were detected when AB8 or AB8-TMPRSS2 podocytes were inoculated with the pseudoviruses, yet infected cells could be detected when ACE2 was present. Cell susceptibility to viral entry was only slightly increased in cells that co-expressed TMPRSS2 only in the case of the Delta variant, similar to what we observed when cells were infected with SARS-CoV-2. These results are in strong contrast to what is observed in the human lung cell line A549. In this commonly used cell model, susceptibility to viral entry was also strictly dependent on ACE2; however, co-expression of TMPRSS2 greatly increased cellular infection (see Appendix A).

The study of SARS-CoV-2 staining in autoptic kidneys from patients who died under COVID-19 showed that when the virus could be detected, it was present mainly in tubular structures (Figure 8).

### 3.3. ACE2 and TMPRSS2 Protein Abundance in Human Kidneys and Colon

Since sex and age are important factors influencing the severity of COVID-19 course, the protein levels of ACE2 and TMPRSS2 in membrane fractions obtained from non-COVID-19 human kidneys and colon samples were examined and analyzed as a function of sex (Figure 9) and age (Figure 10). Interestingly, in contrast to colon samples, in kidney samples a dependence of ACE2 protein level on sex was detected: kidneys from female patients showed a significantly higher ACE2 protein abundance in the membrane fraction than kidneys from male patients (Figure 9). Furthermore, in renal samples from female patients, a significant inverse correlation between ACE2 level and age was detected (Figure 10). In colon samples, no significant difference in ACE2 protein amount between female and male patients and no age-dependence was observed (Figure 9). Protein abundance of TMPRSS2 could be determined only in some kidney samples, as in most of the specimens it was underneath the detection threshold. In these renal samples, the TMPRSS2 protein level was also low and not sex-dependent (0.09 ± 0.01 pmol/mg protein, N = 5 and 0.07± 0.01 pmol/mg protein, N = 3, in kidney samples from male and female patients, respectively; *p* = 0.4, unpaired *t*-test). The protein abundance of Na^+^/K^+^-ATPase in kidney samples from male and female patients, used as controls for sample preparation, was not significantly different (4.9 ± 0.5 pmol/mg protein, N = 9, and 6.2 ± 0.5 pmol/mg protein, N = 10, respectively; *p* = 0.1, unpaired *t*-test).

## 4. Discussion

In addition to the lungs, human kidneys are reported to be frequently affected by SARS-CoV-2. As evidence of renal involvement, computed tomography scans from COVID-19 patients showed signs of inflammation and edema within the renal parenchyma [51]. Kidney dysfunctions including haematuria, proteinuria, or increased levels of serum creatinine, uric acid, or blood urea nitrogen are linked to poor prognosis.

As a cause of renal involvement, a possible direct viral cell entry into human kidney cells is largely discussed. Studies reported the detection of viral RNA and protein or virus-like particles in podocytes and tubular epithelium as a proof of a direct viral infection [52,53,54]. However, these observations could not be confirmed by other groups, and, moreover, it was proposed that other cell components such as the endoplasmic reticulum or clathrin-encoded vesicles were misinterpreted as viral particles in the above-cited electron microscopy studies [55,56,57,58]. Here, we found that in kidneys of patients who died from COVID-19, SARS-CoV-2 nucleocapsid staining was detected mainly in renal tubules, confirming the possibility of direct infection of renal cells by SARS-CoV-2.

Since both ACE2 and TMPRSS2 are described as the major cell entry receptor and cellular S-protein priming protease for SARS-CoV-2 on the surface of target cells, respectively [8], a simultaneous expression of both is expected to successfully drive the viral cell entry. However, we have found that the presence of ACE2 is necessary and sufficient to cause SARS-CoV-2 infection of renal cells such as AB8 podocytes. Even though the investigation of ACE2 and TMPRSS2 in human kidney tissues indicates a co-expression of ACE2 and TMPRSS2 in the apical membrane domain of proximal tubules, TMPRSS2 protein amount was only detectable by LC-MS/MS in few kidney samples and was not detected by Western blot analysis of human kidneys (Appendix A), suggesting a proportionally higher ACE2 expression compared with that of TMPRSS2. Accordingly, if kidney cells are a target for SARS-CoV-2 infection, ACE2 probably plays a prominent role in this process. A disbalanced expression ratio of ACE2 and TMPRSS2 was also described in other tissues that are speculated to be affected by SARS-CoV-2. For example, reports of new-onset diabetes in patients suffering from COVID-19 led to the speculation that the pancreas is a possible target for SARS-CoV-2, yet a co-expression of ACE2 and TMPRSS2 was not found within pancreatic ß-cells, and ACE2 had a broader tissue expression than TMPRSS2 [59]. Nevertheless, in other tissues cell entry depending on the function of ACE2 and TMPRSS2 is still probable as single cell RNA-sequencing data revealed a co-expression of both in several cell types, including ileal absorptive enterocytes, nasal goblet secretory cells, and lung type II pneumocytes [60]. In conclusion, a cell entry of SARS-CoV-2 driven by the coupled action of ACE2 and TMPRSS2 might be prominent in some tissues, including human lungs, whereas based on a missing co-expression or disbalanced expression ratio, this cell entry mechanism is improbable in other tissues, such as human kidneys. This points at ACE2 as the crucial component of the viral SARS-CoV-2 cell entry machinery, whereas TMPRSS2 appeared to be dispensable, at least in the kidneys. Indeed, using a human lung cell line, we could confirm the importance of ACE2 for virus infection; however, here an overexpression of TMPRSS2 greatly increased virus infectivity.

A high amount of ACE2 was observed mainly in the apical membrane domain of proximal tubules and we detected SARS-CoV-2 mainly in renal tubular structures corresponding to these proximal tubules, probably representing the virus entry port in the kidneys. In this case, however, SARS-CoV-2 has first to pass the glomerular filtration barrier to reach its receptor. CoVs have a variable size of 80–120 nm in diameter and, therefore, they are probably too big to penetrate across the ∼25–60 nm-wide podocyte slit barrier [61], which constitutes the main filtration barrier in the kidneys. Therefore, it can be supposed that patients with existing defects of the glomerular filtration barrier, which are characteristics of several renal pathologies, are more susceptible to renal SARS-CoV-2 infection during COVID-19. Indeed, pre-existing dysfunctions were observed to possibly worsen and develop to an AKI under COVID-19 infection, which is also linked to higher fatality rates [22,51]. In total, a 5.3-fold higher mortality risk in COVID-19 patients suffering from AKI was described [51].

Interestingly, we found that in kidney samples from female patients, the protein abundance of ACE2 was higher than in kidneys from male patients. Otherwise, ACE2 protein abundance in the plasma membrane from female kidneys decreases with age. In samples from human colon, ACE2 protein abundance was neither dependent on sex nor on age of the patients. The ACE2-coding gene is located on the X-chromosome [24]. Normally, the expression of genes encoded by one of the two X-chromosomes in mammalian females is repressed by X-chromosome inactivation. However, in some tissue, the repression of X-chromosome-located genes can be incomplete, causing their sex-biased expression [62]. We suppose that X-chromosome inactivation in kidneys is not complete, which would explain elevated ACE2 levels in females. Age-related expression patterns of ACE2 within the human kidneys may further depend on the expression of sex hormones. In this line, an upregulation of ACE2 mRNA was reported to be mediated by estrogens [63]. A protecting function of estrogen receptor signaling against SARS-CoV was already described in mice experiments [64]. Downregulation of estrogen production in older females would therefore explain reduction of ACE2 levels in the kidneys. Therefore, decreasing ACE2 levels may increase the activity of the deleterious arm of RAS signaling and with this the possibility to develop a more severe COVID-19 course. Moreover, sex hormones are reported to influence the immune response [65]. Thus, modulation of the immune response by androgens may favor the observed severity of COVID-19 in male patients.

Putting our data into the context of severe COVID-19 risk groups, higher ACE2 protein abundances in kidney samples were found in female patients, who are less likely to undergo a severe course of infection. In hand with our results, other studies found a correlation of lower ACE2 expression and higher COVID-19 severity. By genotype-tissue expression analysis in 30 tissues across thousands of individuals, Chen et al. reported a decreasing ACE2 expression at older ages in many tissues. Moreover, they found a significant downregulation of ACE2 mRNA in pancreatic ductal cells from patients with type II diabetes, a comorbidity that is classified as a risk factor in COVID-19 [63].

Our results suggest a potential protecting function of ACE2 in COVID-19 which may derive from its role in the RAS. Endocytosis of ACE2 after binding of the S-protein of SARS-CoV-2 decreases ACE2 availability and function. A generally lower level of ACE2 would quickly lead to an imbalance of the RAS. This imbalance favors the Ang II/AT1R axis of the RAS, thus triggering deleterious events including inflammation and fibrosis, and in the kidneys may be a factor implied in the observed relapse of glomerular disease after COVID-19 vaccination [66]. In general, an imbalanced RAS is linked to pathological conditions, thus possibly enabling complications of COVID-19, including coagulation or CRS. Therefore, even if the ACE2 level is reduced by the binding of S-protein, a higher presence of ACE2 in female patients could still ensure an adequate ACE2 level, limiting the action of the Ang II/AT1R axis.

In mice models, an influence of the RAS components ACE and ACE2 was already described in severe acute lung injury provoked by sepsis or acid aspiration. ACE was found to promote lung injury, whereas ACE2 attenuated it [67].

Another possible protective function of a higher ACE2 level can be hypothesized if it is assumed that a higher proportion of membrane-bound ACE2 corresponds to higher formation of soluble ACE2 (sACE2). It is conceivable that circulating sACE2 binds the viral S-protein, preventing subsequent binding and internalization of the virus into target cells. The soluble form of the cell entry receptor may neutralize the virus within the body and thereby decrease its infectivity. Indeed, decreasing infectivity of SARS-CoV-2 under higher availability of sACE2 was already shown in cell culture experiments [68,69,70].

Moreover, a negative correlation of immune signatures (interferon response, B cells, and natural killer cells) and ACE2 in female lungs and lung tissues of younger persons was found, while the correlation was positive in males as well as older people. The positive correlation of immune signatures and ACE2 expression in males and the elderly might serve as a potential explanation for the higher case severity of COVID-19 in these people [71]. Thereby, stronger immune signatures may favor an excessive immune response, including, for example, CRS.

Taken together, our results, in hand with other studies, highlight the importance of ACE2 in the cell entry of SARS-CoV-2 [8,9], whereas, in contrast to previous reports, the priming protease TMPRSS2 was found as dispensable in the infection of cultured AB8 podocytes. Moreover, TMPRSS2 protein expression was nearly absent in human colons and low in kidney tissues of both sexes. In consequence, viral cell entry into tissues without or with insufficient TMPRSS2 expression might be driven by alternative S-protein priming processes. In this line, other serine proteases are reported to be involved in infection by respiratory viruses [72,73,74].

This study confirmed ACE2 as an important cell entry receptor of SARS-CoV-2. Moreover, minor importance of the serine protease TMPRSS2 for viral infection of AB8 podocytes in cell culture experiments was demonstrated. Additionally, age- and sex-related differences in the protein abundances of the virus cell entry receptor ACE2 in human kidneys and direct virus infection of renal tubules were demonstrated.

## 5. Conclusions

In conclusion, this work highlights the assumption that observed renal pathogenesis during COVID-19 results from multifactorial causes. These causes might include a direct viral entry into renal cells, which leads to disruptions and local damages in regions where the virus replicates. Moreover, indirect causes such as an imbalance of the RAS can cause secondary infection events including fibrosis or inflammation which also negatively impact kidney function, and the occurrence of CRS can furthermore promote inflammatory processes and damages and may result in organ failure. Thereby, an interplay of these and further factors can be the cause of the widely varying pathogenesis in the human kidney under SARS-CoV-2 infection.

## Figures and Tables

**Figure 1 biomolecules-13-00472-f001:**
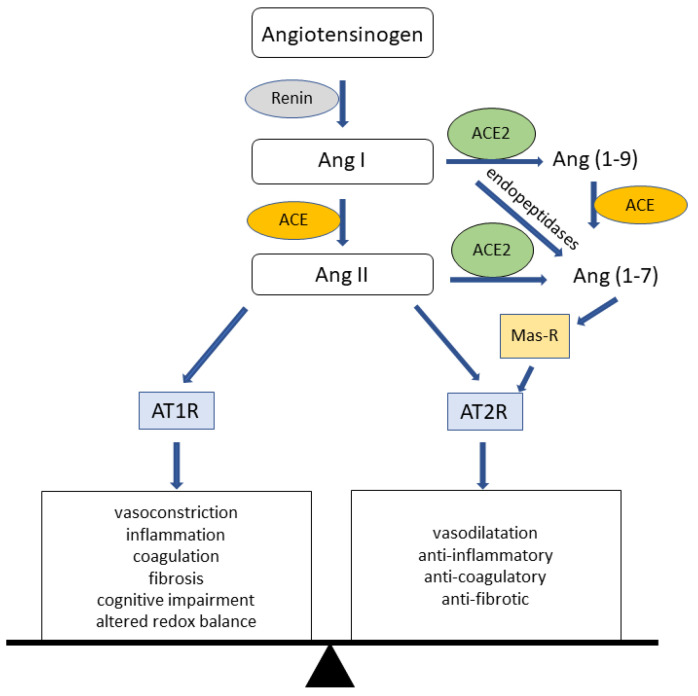
The renin–angiotensin system (RAS). Angiotensinogen is cleaved into Angiotensin I (Ang I) by renin. In turn, Ang I is cleaved to Angiotensin II (Ang II) by angiotensin-converting enzyme (ACE). Ang II is either further cleaved to Angiotensin (1–7) (Ang (1–7)) by angiotensin-converting enzyme 2 (ACE2), which in turn activates Mas-Receptor (Mas-R), or it activates one of the Ang II receptors type 1 or 2 (AT1R or AT2R). These receptors in turn provoke detrimental or protecting effects of RAS signaling (modified from [25]).

**Figure 2 biomolecules-13-00472-f002:**
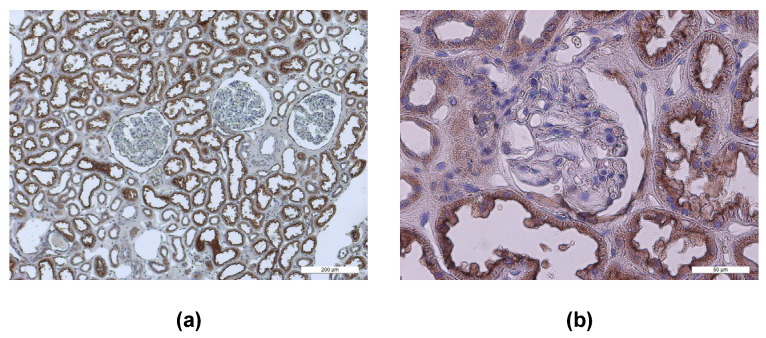
ACE2 staining in human kidneys. Panel (**a**) shows that ACE2 is mainly expressed in the tubular structures in the kidney cortex, whereas glomeruli seem not to express ACE2. Scale bar indicates 200 µm. Panel (**b**) shows a photo taken at higher magnification, where again an intensive ACE2 staining was observed in the apical membrane domain of renal tubules, whereas the glomerulus in the center of the photo shows no or a very low signal for ACE2. Scale bar indicates 50 µm.

**Figure 3 biomolecules-13-00472-f003:**
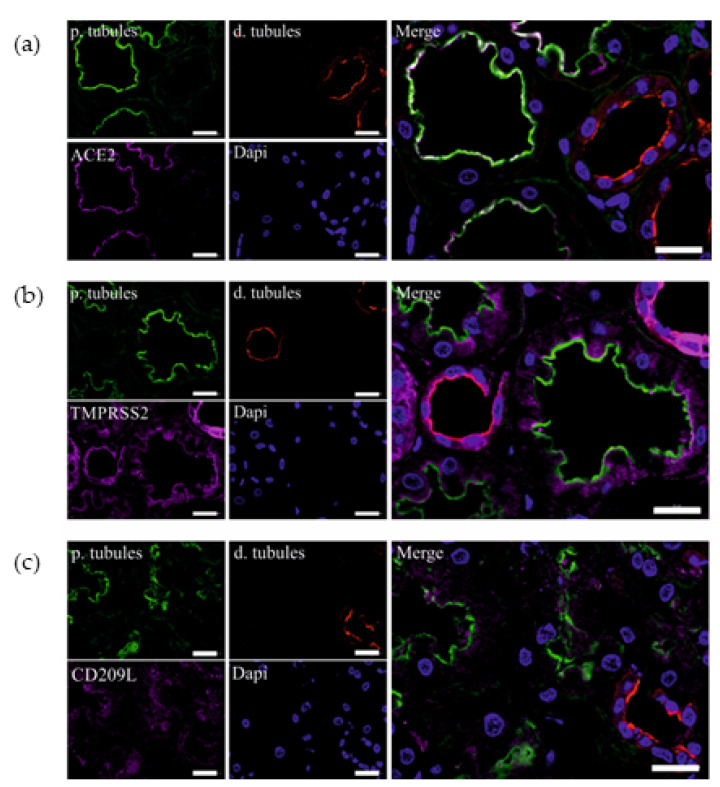
ACE2 (**a**), TMPRSS2 (**b**), and CD209L (**c**) expression in human kidney tissue. Proximal tubuli (p. tubules) were stained by PHE-A (green) and distal tubules (d. tubules) by PNA (red). Nuclei were stained with DAPI (blue). Panel (**a**) shows an immunofluorescence analysis of ACE2 expression in human kidney tissues. As evident from examination of the merge picture, ACE2 signal (pink) co-stained with green staining of PHE-A, originating a white color, indicating ACE2 localization on the apical membrane domain of cells from the proximal tubules. No expression of ACE2 in other nephron segments was observed. Panel (**b**) shows an immunofluorescence analysis of TMPRSS2 expression (pink) in human kidney tissues. As evident from examination of the merge picture, the TMPRSS2 signal was detectable both in renal proximal and distal tubules, which were labeled with PHE-A (green) and PNA (red), respectively. Panel (**c**) shows an immunofluorescence analysis of CD209L expression (pink) in human kidney tissues. As evident from examination of the merge picture, CD209L signal was weak and mainly evident in cells from the proximal tubules, which were labeled with PHE-A (green). Scale bars indicate 20 µm.

**Figure 4 biomolecules-13-00472-f004:**
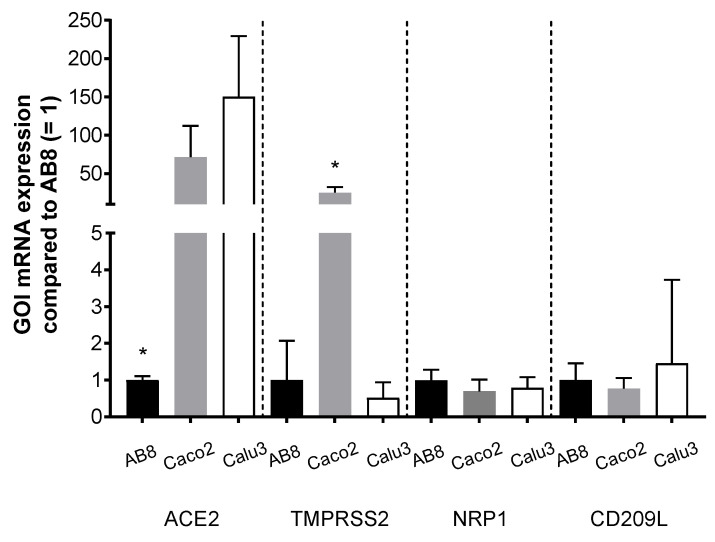
Endogenous expression of mRNA (mean ± SEM) for putative SARS-CoV-2 cell entry factors (genes of interest (GOI): ACE2, TMPRSS2, NRP1, and CD209L) in human immortalized podocytes (AB8), Caco2-, and Calu3 cells. The expression of mRNA in AB8 was set to 1. The Ct values for ACE2, TMPRSS2, NRP1, CD209L, and GAPDH in AB8 podocytes were 30.5 ± 0.1, 27.4 ± 1.0, 23.3 ± 1.3, 29.1 ± 0.4, and 15.1 ± 0.6, respectively; N = 3. Asterisks indicate a statistically significant difference compared to the other cell line (Anova with Tukey’s multiple comparisons test, *p* < 0.05).

**Figure 5 biomolecules-13-00472-f005:**
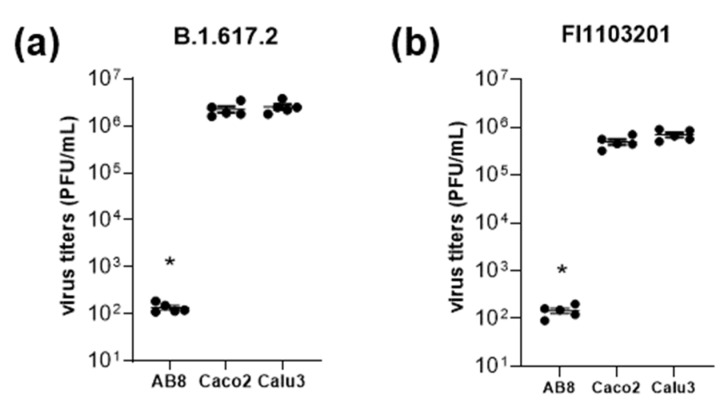
SARS-CoV-2 infectivity of the variants B.1.617.2 (panel **a**) and FI1103201 (panel **b**) in AB8, Caco2, and Calu3 cells. Cells were infected with the SARS-CoV-2 variants and infection rates were calculated by a plaque assay. Thereby, AB8 cells displayed a low infectivity rate, while Caco2 and Calu3 cells were efficiently infected with both variants. Single data points from 5 independent experiments and mean ± SEM are shown. Asterisks indicate a statistically significant difference compared to Caco2 and Calu3 cells (Anova with Dunnett’s multiple comparison test, *p* < 0.05).

**Figure 6 biomolecules-13-00472-f006:**
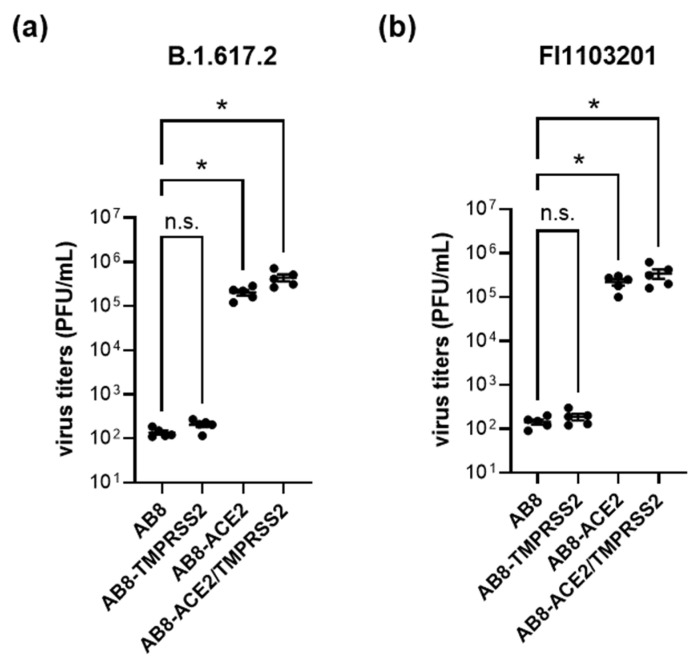
Overexpression of ACE2 increases SARS-CoV-2 infectivity of the variants B.1.617.2 (panel **a**) and FI1103201 (panel **b**). Cells were infected with the two SARS-CoV-2 variants and infection rates were calculated by a plaque assay. Thereby, AB8 cells displayed a low infectivity rate, which was not enhanced by overexpression of TMPRSS2 (AB8-TMPRSS2) but was strongly increased in AB8-ACE2 and AB8-ACE2/TMPRSS2 cells. Single data points from 5 independent experiments and mean ± SEM are shown. Asterisks indicate a statistically significant difference compared to AB8 human podocytes (*p* < 0.05, Anova with Dunnett’s multiple comparison test); ns = not significant.

**Figure 7 biomolecules-13-00472-f007:**
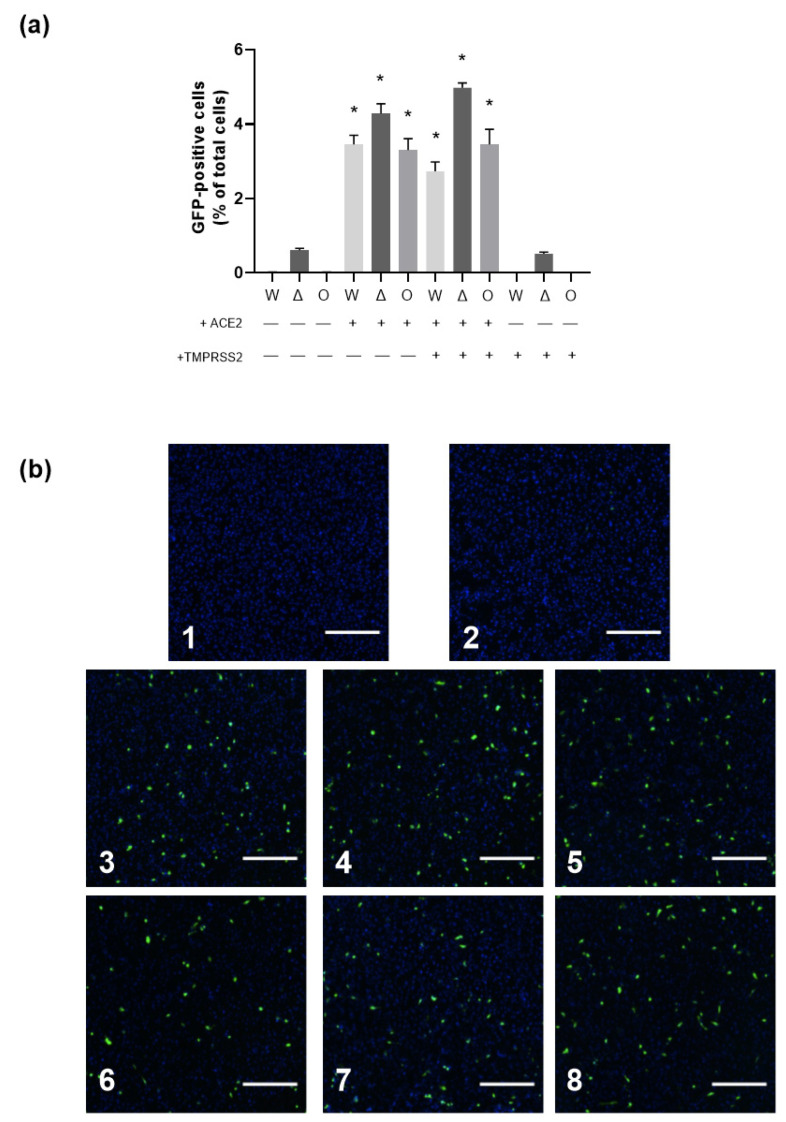
Dependence of SARS-CoV-2 pseudovirus entry on ACE2 and TMPRSS2 expression in AB8 podocytes. Wildtype AB8 podocytes (WT) and AB8 podocytes expressing either ACE2 (+ ACE2), TMPRSS2 (+TMPRSS2) or both (+ ACE2 + TMPRSS2) were inoculated with VSVΔG/GFP-SARS-CoV-2 spike particles bearing Wuhan (W), Delta (Δ), or Omicron (O, BA.5) spike variants. GFP-positive cells were quantified. Panel (**a**) Quantitative analysis of GFP-positive cells. Data are given as mean ± SEM. The numbers of independent experiments were 5 everywhere. * shows a statistically significant difference to all the groups without ACE2 overexpression (*p* < 0.01, Anova with Tukey’s multiple comparison test). Panel (**b**) shows exemplary fluorescence images. Cells were detected by staining the nuclei with Hoechst and were automatically counted. GFP-positive cells were automatically detected. No or very low pseudovirus entry was detected in AB8 WT podocytes (panel (**b1**)) and after overexpression of TMPRSS2 alone (panel (**b2**)). Overexpression of ACE2 resulted in a well-measurable number of GFP-positive cells (panels (**b3**–**b5**) for the variants W, Δ, and O, respectively), which was only slightly increased in the case of the Δ variant by the concomitant presence of TMPRSS2 (panels (**b6**–**b8**) for the variants W, Δ, and O, respectively). Scale bars indicate 500 µm.

**Figure 8 biomolecules-13-00472-f008:**
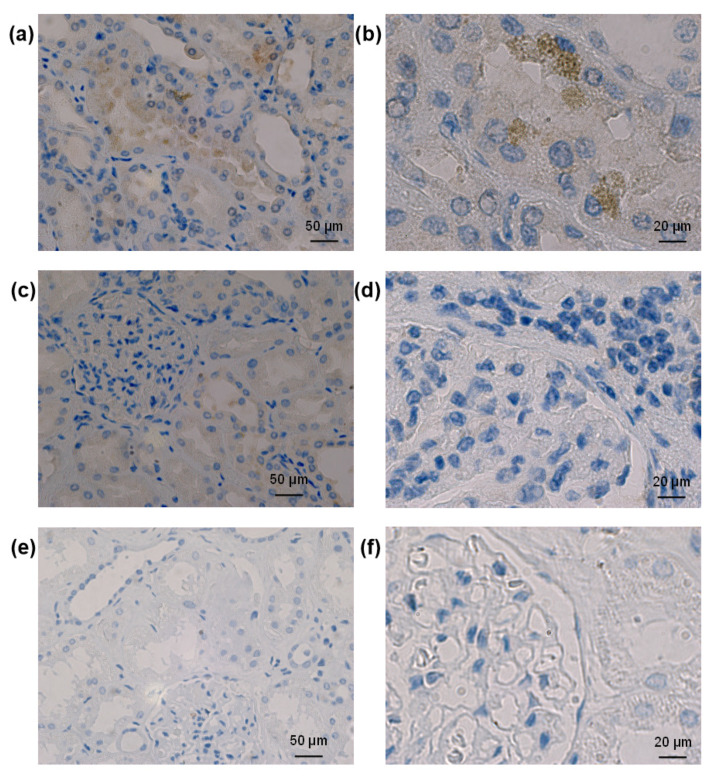
Analysis of SARS-CoV-2 nucleocapsid staining in human kidneys. A staining for SARS-CoV-2 could be detected only in tubular structures (panels (**a**,**b**)) but not in glomeruli (panels (**c**,**d**)) of some autoptic kidneys from patients who died from COVID-19. Staining with the antibody against SARS-CoV-2 in non-COVID-19 human kidneys (panels (**e**,**f**)) resulted in no signal, confirming the specificity of the antibody used.

**Figure 9 biomolecules-13-00472-f009:**
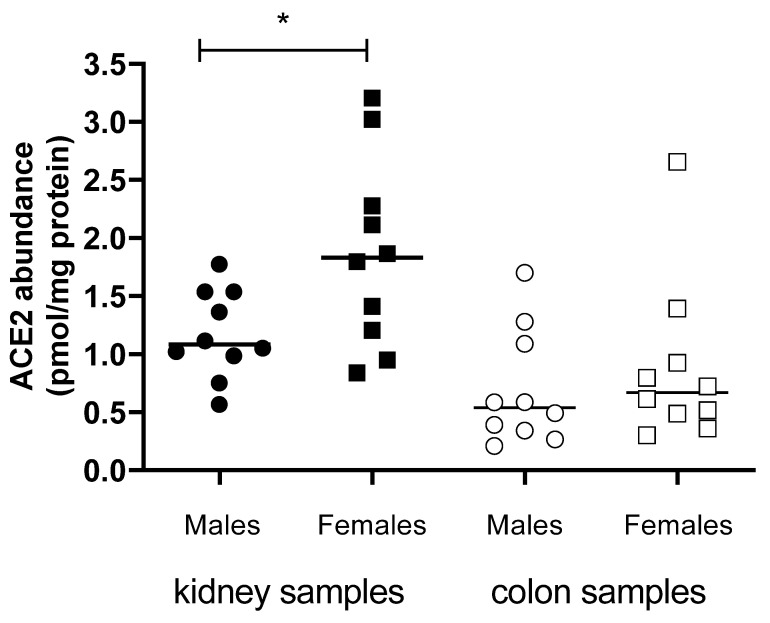
ACE2 protein abundance (pmol/mg membrane protein) in the plasma membrane of samples from human kidney (closed symbols) and colon (open symbols) samples. The bar represents the median value. The asterisk indicates a statistically significant difference between ACE2 protein abundance in kidney samples from female and male patients (*p* < 0.05, unpaired *t*-test).

**Figure 10 biomolecules-13-00472-f010:**
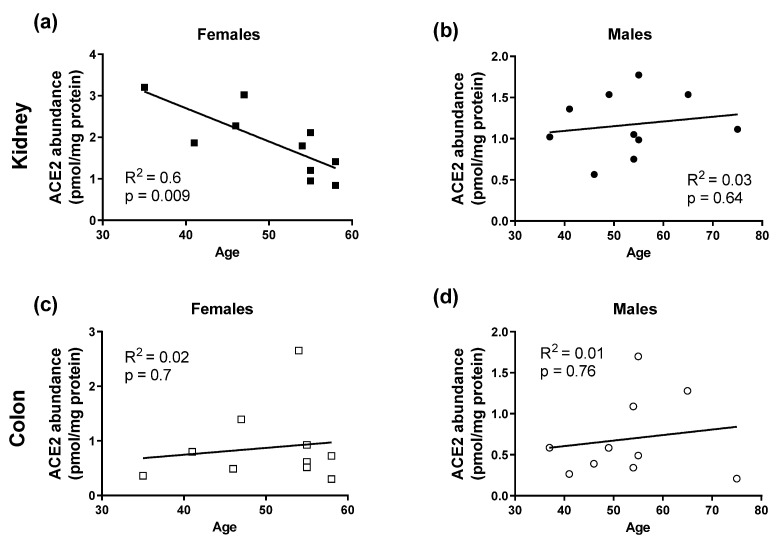
The relationship between ACE2 protein abundance (pmol/mg membrane protein) in kidney samples (panels (**a**,**b**), closed symbols) and colon samples (panels (**c**,**d**), open symbols), obtained from both female (squares) and male (dots) patients, and their age. Shown are linear regression lines (solid lines) along with the regression coefficients (R^2^) and *p* values of the test, showing whether the regression line is significantly different from zero.

**Table 1 biomolecules-13-00472-t001:** Oligonucleotides used for real-time PCR analysis.

Primer	Sequence 5′–3′
ACE2 sense	CATTGGTCTTCTGTCACCCGA
ACE2 antisense	CCCCAACTATCTCTCGCTTCATCT
TMPRSS2 sense	AAATCCCCATCCGGGACAGT
TMPRSS2 antisense	GAGCACTTGCTGCCCATGAA
CD209 ^1^ sense	TTCCAGAAGTAACCGCTTCACC
CD209 antisense	CTGCTTGAAGCTGGGCAACA
CD209L ^2^ sense	AGTGGCTGGAACGACAATCG
CD209L antisense	AGAGAACCGTCTCAGGGCAT
NRP1 ^3^ sense	AGGATCACACAGGAGATGGC
NRP1 antisense	CTGGTAGCGCAGTTTGACCC

^1^ CD209 = Dendritic cell-specific ICAM-3-grabbing non-integrin 1 (DC-sign); ^2^ CD209L = Liver/lymph node-specific ICAM-3-grabbing non-integrin (L-sign) [12]; ^3^ NRP1 = neuropilin-1 [11].

## Data Availability

Not applicable.

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
