# Peer review of "Importance of ACE2 for SARS-CoV-2 Infection of Kidney Cells"

_biomolecules, 2023, doi:10.3390/biom13030472_

Round 1
Reviewer 1 Report
The aim of current study by Kroll et al is to analyze the ACE2 expression, importance and viral entry in kidney cells. The data presented is very interesting and sound. I have following points
1. ACE2 expression by various cell types and possible entry of SARS-CoV-2 is long debated where majority of data highlights over expression of ACE2 protein by cells play a positive role, however, data shown does not support this specially with kidney cells. Authors used AB8 podocytes, Caco2 and Calu3 lines, where Caco2 and Calu3 are over expressing ACE2, how, authors can explain this in context of their emphasis on kidney infection during COVID-19.
2. Can authors show some other negative or low ACE2 expressing cells in body to highlight its importance where kidney showing highest expression (e.g. skin cells).
3. Data regarding ACE2 expression from samples is interesting, however, how such data (Females kidneys having higher ACE2?) can be related with COVID infection as gender biased.
4. Some linguistic errors should be omitted e.g. line 376, 379.
Author Response
"Please see the attachment."

Reviewer 2 Report
In this manuscript, the authors aimed to investigate the role of ACE2 and TMPRSS2 in SARS-CoV-2 infection of human kidneys. They showed that ACE2 protein was highly expressed in human kidney cells, especially the apical membrane domain of renal tubule cells, while TMPRSS2 was only detected in a few kidney samples by mass spectrometry. They concluded that ACE2 is necessary and sufficient for SARS-CoV-2 infection of human kidneys, however, TMPRSS2 seems dispensable in this process. This observation is interesting and novel, however, the evidence about how the expression profile of ACE2 in kidneys correlates the COVID-19 symptoms in female and the aged populations was weak. Therefore, the major concerns below should be addressed before acceptance to publish in the Biomolecules.
Major points:
1. In Figures 5 and 6, it will be strong evidence if the authors can present the viral infection rate by different assays like IF and WB shown in (https://doi.org/10.3390/biomedicines9091230; https://doi.org/10.1101/2022.03.04.483074)
2. Western blotting results should be shown in Figures 8 and 9 to support the conclusion that ACE2 was expressed higher in female kidney samples. Also, in line 433, the authors should explain or comment on why kidneys from female patients showed a significantly higher ACE2 abundance and how it correlates to the fact that the rate of severely affected and hospitalized patients is significantly higher in males.
3. It is confusing for lines 534 and 535, to say that reduction of ACE2 levels and more severe COVID-19 infections could be due to decreased expression of estrogen in old females. Please check and make it more clear.
4. It will be good if authors can show the protein expression and localization by WB and IF of stable cell lines in Figure 6.
Minor points
1. In line 202, TMRPSS2 should be TMPRSS2; in line 226, Triton-X-100 should be Triton-X 100; in line 228, please check whether the washing solution includes 0.2% Triton-X 100, if yes, how to guarantee complete washout of Triton-X 100 in line 227? In lines 235-236, DAPI was incubated for 1 hour, is it true? In line 380, express also should be also express.
2. In Figure 2, please add the scale bar in the legend.
3. Both Caco2 and CaCo2 were used in the manuscript. Check and keep consistent.
Author Response
"Please see the attachment."

Reviewer 3 Report
Major Comments
Discussion: please mention that by this time the majority of the worlds population has taken one or more COVID-19 vaccines, off of which expose the human body to the Spike protein. Thus SARS-CoV-2 infection in a vaccinate person poses enhanced risk via the mechanisms proposed. Please cite the literature on COVID-19 vaccination, kidney injury, progression of CKD in glomerulonepathies (Canney et al JASN).
Author Response
"Please see the attachment."

Round 2
Reviewer 2 Report
In this revised manuscript, the authors performed new experiments including pseudovirus entry assay to specifically focus on the viral entry step and immunofluorescence imaging to analyze the ACE2 and TMPRSS2 localization, which strongly supports their conclusion and greatly improved the quality of the manuscript. However, there are a few problems that should be addressed before acceptance for publication.
First and most important one, in the new Figure 7a, the best percentage of infected cells in the total is less than 1%, is it true? If yes, that is too low. In Figure 7b, the GFP-positive cells are too few in panels 3-5 and 6-8, which suggests the quality of the results is poor and cannot support their conclusion. Authors need to improve the pseudovirus quality or add more viruses to enhance the infection rate. It will be good to make the infection rate of AB8 cells similar to A549 in supplemental figure 4.
Second, in supplemental Figure S3, the TMPRSS2 localization is wrong. Please try more antibodies of TMPRSS2 if possible. The nuclear localization of TMPRSS2 had better be removed if no new antibodies work.
Also, to comment on the answers to my second question below, the running out of samples could be an excuse, but the validation of your mass spectrometry results by western blotting or other methods will never be outdated because it is always good to support your conclusion by different methods and your results should support each other, which will help to prevent or reduce the possible misunderstanding or misinterpretation. Also, western blotting is generally more sensitive compared to mass spectrometry.
“2. Western blotting results should be shown in Figures 8 and 9 to support the
conclusion that ACE2 was expressed higher in female kidney samples. Also, in line
433, the authors should explain or comment on why kidneys from female patients
showed a significantly higher ACE2 abundance and how it correlates to the fact that
the rate of severely affected and hospitalized patients is significantly higher in males.
Answer: Unfortunately, we have used all the available tissue for mass-spectrometry
(MS) analysis, and, therefore, are not able to perform the requested experiments.
Moreover, authors who submit papers containing quantitative protein data generated
via MS are frequently asked by reviewers to validate some of the values with Western
blotting. We believe that with the advances that have occurred, this request is now
outdated, causing the unnecessary use of scarce resources and not achieving the main
intent: objective cross-validation of results. The quality of MS data is vastly superior to
that of Western blot analysis for several reasons, which are well summarized in Mol
Cell Proteomics. 2013;12(9):2381-2. doi: 10.1074/mcp.E113.031658.
